# Predictive Value and Therapeutic Significance of Somatic BRCA Mutation in Solid Tumors

**DOI:** 10.3390/biomedicines12030593

**Published:** 2024-03-06

**Authors:** Gyongyver Szentmartoni, Dorottya Mühl, Renata Csanda, Attila Marcell Szasz, Zoltan Herold, Magdolna Dank

**Affiliations:** Division of Oncology, Department of Internal Medicine and Oncology, Semmelweis University, 1083 Budapest, Hungary

**Keywords:** somatic *BRCA1/2*, PARP inhibition, breast neoplasms, ovarian neoplasms, pancreatic neoplasms, prostatic neoplasms

## Abstract

Ten percent of patients with breast cancer, and probably somewhat more in patients with ovarian cancer, have inherited germline DNA mutations in the breast and ovarian cancer genes *BRCA1* and *BRCA2*. In the remaining cases, the disease is caused by acquired somatic genetic and epigenetic alterations. Targeted therapeutic agents, such as poly ADP-ribose polymerases (PARP) inhibitors (PARPi), have emerged in treating cancers associated with germline *BRCA* mutations since 2014. The first PARPi was FDA-approved initially for ovarian cancer patients with germline *BRCA* mutations. Deleterious variants in the *BRCA1/BRCA2* genes and homologous recombination deficiency status have been strong predictors of response to PARPi in a few solid tumors since then. However, the relevance of somatic *BRCA* mutations is less clear. Somatic *BRCA*-mutated tumors might also respond to this new class of therapeutics. Although the related literature is often controversial, recently published case reports and/or randomized studies demonstrated the effectiveness of PARPi in treating patients with somatic *BRCA* mutations. The aim of this review is to summarize the predictive role of somatic *BRCA* mutations and to provide further assistance for clinicians with the identification of patients who could potentially benefit from PARPi.

## 1. Introduction

The discovery of the BReast CAncer (*BRCA*) genes in the early 1990s, namely *BRCA1* and *BRCA2*, may be considered as one of the most significant achievements in medicine lately. It was found that both genes play a crucial role in the development of hereditary breast cancer (BC). Moreover, it soon became clear that these genes are involved in the pathogenesis of familial ovarian cancer. *BRCA1* and/or *BRCA2* gene mutations have since been shown to play a role in many more tumor types [1,2,3]. *BRCA1* was discovered in 1994, and it is located on chromosome 17q21 containing 22 exons. *BRCA2* was discovered a year later, in 1995, contains 27 exons, and it is located on chromosome 13q12 [2]. Of the two transcriptomes, the BRCA2 protein is the larger, which is mostly involved in homologous recombination (HR). The BRCA1 protein has more active functions, which are carried out through various functional domains of the molecule that can interact with a range of other proteins. Its involvement in DNA repair, checkpoint control of the cell cycle, protein ubiquitination, and chromatin remodeling were previously described [2,4].

The *BRCA1* and *BRCA2* genes are indirect tumor suppressor stability genes [3]. Both germline (g*BRCA*) and somatic (s*BRCA*) mutations of the *BRCA* genes are known (Figure 1). A large number of *BRCA* mutation variants are known to significantly increase the risk of developing certain cancers. Most data are available for breast, ovarian, pancreatic, and prostate tumors, but other tumors are also known to be affected [1]. For example, the involvement of *BRCA* genes was reported for cholangiocellular cancer [5,6], melanoma [7], bladder cancer [8], non-small cell lung cancer [9], and gastrointestinal tumors, including esophageal, gastric, and colorectal cancers [10,11,12,13]. The incidence of *BRCA* mutations within the general population is 1:300–800 [14]. Worldwide, 5% to 10% of breast cancers (BCs), 10% to 15% of ovarian cancers, 4% to 7% of pancreatic cancers, 6% of patients with metastatic prostate cancer, and rarely other cancer-type cases originated from those carrying germline *BRCA1/2* (g*BRCA1/2*) mutations [9]. In Hungary, approximately 4–10% of breast cancers, 11% of ovarian cancers, and 33% of male breast cancers are *BRCA* mutant [15,16,17]. The most common *BRCA* mutation variants of European countries are summarized in the article by Dr. Janavičius [18]. The lifetime risk of developing cancer in g*BRCA1/2* carriers is very high, up to 70–80%, compared to that of the much lower risk in the general population [19]. Both *BRCA* genes are autosomal dominant; therefore, the offspring has a 50% probability of inheriting the mutated variant [3]. g*BRCA* is (usually) detected in blood, whereas somatic *BRCA* (s*BRCA*) mutations could be identified from genomic profiling of tumor tissue or by testing the circulating tumor DNA [19].

*BRCA1/2* mutant tumors respond well to the treatment with poly (ADP-ribose) polymerase (PARP) inhibitor (PARPi) treatments. PARPis are novel drugs acting on the base excision repair pathway. In HR-deficient cells, such as the ones with the *BRCA1/2* mutations, due to the blockage, the DNA double-strand breaks cannot be repaired, which ultimately results in apoptosis [20,21]. The use of PARPi in *BRCA*-related malignancy has largely been limited to g*BRCA* mutations by most guidelines. Targeted treatment of somatic mutations is suggested only in ovarian [22] and prostate [23] cancers, while the precise estimate of the efficacy of PARPi in somatic *BRCA* mutation is still lacking. In other cancer types, the current clinical data is scarce [5], *BRCA* studies are still ongoing [13], or somatic *BRCA* mutations are under-recognized and represent a missed opportunity for PARPi-targeted therapy. It has also to be mentioned that many of the existing clinical trials do not specifically include somatic *BRCA* patients, leading to underrepresentation in the data.

As detailed in the previous paragraph, clinical and treatment data on s*BRCA* is scarce, even for common cancers like breast, ovarian, pancreatic, and prostatic cancer. To our knowledge, this is the first review on this subject, where the main goal is to synthesize the available clinical and treatment knowledge on breast, ovarian, pancreatic, and prostatic cancers with s*BRCA* mutations.

## 2. Clinical Differences of s*BRCA* and g*BRCA*

Cancers caused by pathogenic g*BRCA* variants are hereditary and can be passed down over generations. These mutations are constant, and there are certain patterns, like positive family history and well-known cancer syndromes. The pathogenic variant is discovered by testing peripheral blood samples for specific, known mutations in the *BRCA* genes. Usually, screening methods are developed for early diagnosis, and the clinical management of family members is highly recommended. In these cases, extensive genetic counseling is necessary, not just the oncological care of the given patient. In contrast, somatic mutations are detected in the tumor tissue or by the analysis of circulating tumor DNA. Cancers caused by s*BRCA* mutation(s) are sporadic. These are not found in every cell in the body, and they cannot be inherited but rather caused by some external noxa (virus, chemical exposure, etc.) or aging. However, in both cases, testing may help to search for optimal treatment selection (Table 1) [2,24].

Germline testing should be considered for patients with ovarian, breast, prostate, and pancreatic cancer if certain risk factors are present, e.g., positive family history, bilateral disease, multiple primary tumors, and/or young age at the onset of tumor diagnosis. In case of positive test results, the testing of the family members should be considered. Nowadays, genetic counseling for affected individuals is recommended by guidelines, involving a genetic expert [24,25].

The use of multigene profiling of the tumor tissue in today’s oncology practice is increasingly recommended and enables targeted therapy, mainly in the metastatic setting. However, there are certain differences between germline and somatic testing and findings. The interpretation of the results of multigene tests and the role and importance of variants are not entirely explored [19]. The prognostic and predictive roles of g*BRCA* mutations have been largely demonstrated and shared in the last two decades. It is not entirely clear whether harboring s*BRCA* mutation(s) detected by the analysis of the tumor tissue brings the same prognostic and predictive advantages and whether there is a possible targeted treatment. There are several reasons for this uncertainty. Some tumors do not respond equally to targeted therapies due to differences in biological behavior and environmental interaction. Moreover, challenges in somatic mutation testing can also influence the result, e.g., laboratory expertise, high-quality tissue selection, DNA isolation, and correct variant interpretation. Somatic mutations in patients may change over time according to the site of tumor evaluation. Furthermore, as a result of certain systemic treatments, the genetics of the tumor might also change, including the development of resistance, ultimately leading to a change in the mutational status of the lesion [26].

Molecular profiling is recommended as a standard care in advanced/metastatic tumors. In addition to current practice and frequency of biopsies in metastatic cancer, s*BRCA* mutations are reported in 2% to 5% and 3% of ovarian and BCs, respectively [26]. Over the years, a large number of pathogenic or likely pathogenic *BRCA* mutations have been identified [27]. However, the real prevalence of s*BRCA* in patients with metastatic cancers is unknown because of a probably lower number of analyzed primary and metastatic lesions. For some tumor types, e.g., breast tumors, a biopsy of newly diagnosed metastases is strongly recommended. Moreover, at this point, it is not known whether germline and somatic *BRCA1/2* mutations are biologically equivalent [9,19].

PARPis can inhibit the activity of PARP1, PARP2, and PARP3, a group of proteins closely involved in the repair of single-strand DNA breaks. Lately, it has been suggested that PARPis have effects on macrophages and inflammatory makers [28,29]. Several molecules have been developed for inhibition of the PARP function, but to date, five PARP1/2 inhibitors have received marketing authorization for cancer treatment worldwide. The initial clinical trial leading to the U.S. Food and Drug Administration’s (FDA) approval for Olaparib in advanced ovarian cancer only examined germline *BRCA* mutations [30]. Since then, only in two other tumor types, ovarian and prostate cancer, exists clear recommendations for treating somatic *BRCA* mutated malignant disease. With the increasing number of biopsies and the broader use of next-generation sequencing, newly recognized somatic mutations may also be important in the choice of treatment for other tumor types [31,32].

## 3. Ovarian Cancer

Carriers of a monoallelic g*BRCA1/2* mutation have a greater risk of cancerous disease in their lifetime [33]. The mean cumulative risk of having ovarian cancer with *BRCA1* pathogenic mutation is 40% [95% confidence interval (CI): 35–46%], and 18% (95% CI: 13–23%) for patients carrying *BRCA2* mutations. The EMBRACE prospective study has shown 59% (95% CI: 43–76%) and 16.5% (95% CI: 7.5–34%) for the same risks, respectively [2,33,34]. It has also been suggested that there are some s/g*BRCA* mutant breast cancer types that share several phenotypic and genomic traits with s/g*BRCA* mutant ovarian cancers [35].

The first PARPi, Olaparib, was FDA-approved in December 2014, initially for metastatic ovarian cancer patients with germline *BRCA* mutations, after multiple cycles of chemotherapy [30,36]. Olaparib is a targeted therapy for DNA damage response and DNA repair pathways. Later, Olaparib was also approved for the treatment of ovarian cancers as a first-line maintenance therapy and in combination with bevacizumab [37,38]. In general, ovarian cancer is often diagnosed in an advanced stage, and despite good sensitivity to taxane– and platinum-based chemotherapy combinations, most patients will relapse in a short time. With the introduction of PARPi, the efficacy of complex treatment approaches has risen [39,40,41]. The updated results of the SOLO1 trial, after 7 years of follow-up, demonstrated a very high percent, nearly 70% survival with Olaparib, and half of the patients did not receive further therapy [40]. Moreover, a statistically significant increase in the five-year overall survival (OS) of patients treated with PARPi was observed in the PAOLA1 study that was published last year [41] (Table 2).

High-grade serous ovarian cancer patients (HGSOC) with *BRCA1/2* alterations are candidates to receive PARPis in second-line therapy since the FDA approval after responding to the first-line platinum-based chemotherapy [42]. Most of the *BRCA* alterations are germline pathogenic mutations, but in 30% of the cases, the alterations can only be seen at the somatic level; therefore, HGSOC patients can benefit from tumor tissue DNA testing for s*BRCA* mutations. In general, in 13–23% of the diagnostic histological samples from HGSOC patients, a pathogenic *BRCA1* or *BRCA2* gene variant can be found [43,44]. In the AGO TR 1 study, 6.3% of the included ovarian cancer patients had somatic *BRCA1/2* gene mutations [45], while the FLABRA study showed a higher than estimated rate of tumor *BRCA* mutations in ovarian cancer patients in a Latin American population of 28%, without specifying whether it was at the germline or somatic level [46].

The ORZORA trial further supported the use of maintenance Olaparib in all patients, including s*BRCA*-mutated tumor carriers. This study evaluated the efficacy and safety of maintenance Olaparib in patients with platinum-sensitive relapsed ovarian cancer after ≥ two lines of treatment. Maintenance Olaparib had similar clinical activity in germline and somatic mutations. The activity was also observed in patients with a non-*BRCA* homologous recombination repair (HRR) gene mutation [47] (Table 2). The OLATRA study aims to scan the efficacy of maintenance Olaparib in relapsed ovarian cancer patients with s/g*BRCA* deleterious mutations after first-line platinum-based chemotherapy, at least a 6-month treatment-free interval since the last platinum and receiving trabectinib and pegylated liposomal doxorubicin [48]. There is a study currently in the state of “Recruiting” to investigate the correlation between resistance to PARPi and homologous recombination deficiency (HRD) status in epithelial ovarian cancer patients, testing every patient for *BRCA* status [49]. The NUVOLA study investigates the addition of Olaparib to neoadjuvant platinum-based chemotherapy in HGSOC patients with g/s*BRCA* mutations [50]. The OLALA (or OZM-061) study is collecting data about patients with epithelial ovarian cancer receiving Olaparib in any setting to (among others) identify the ratio of somatic *BRCA* mutations [51].

Recently, more options have become available with the approval of the agents niraparib and rucaparib, complicating the therapeutic decision for the best-personalized treatment approach. The Athena-MONO trial demonstrated that rucaparib is an effective treatment, not only in the recurrence setting but in first-line maintenance as well [52,53]. The benefit was demonstrated independently from *BRCA* and homologous recombination deficiency (HRD) status or surgical outcome. Patients with advanced-stage high-grade ovarian cancer undergoing surgical cytoreduction and responding to first-line platinum-doublet chemotherapy were enrolled in this maintenance study. Rucaparib significantly improved progression-free survival (PFS) versus placebo, regardless of *BRCA* or HRD status. Trial data suggests that rucaparib maintenance therapy provides benefits for responders to first-line chemotherapy patients without strong dependency on *BRCA* or HRD status [53]. However, data from the ARIEL4 study in metastatic disease did not conclude the use of rucaparib monotherapy after two or three lines of chemotherapy in somatic, or germline mutated *BRCA* patients. Based on the results, FDA approval has been withdrawn in this setting [54,55] (Table 2).

The ENGOT-OV16/NOVA study was a randomized, double-blind, placebo-controlled, phase III trial that enrolled 553 patients with platinum-sensitive, recurrent ovarian cancer for evaluating niraparib. Patients were enrolled into independent germline *BRCA*-mutated and non-germline *BRCA*-mutated cohorts and then randomly assigned in a 2:1 ratio to receive niraparib at 300 mg once daily or placebo after standard platinum therapy. Primary results from the study, released in 2016, indicated a statistically significant PFS benefit for the niraparib maintenance arm, compared to that of placebo in the germline *BRCA*-mutated, non-germline *BRCA*-mutated, homologous repair-deficient and in the homologous repair-proficient populations. Long-term analyses of the second PFS beyond the first disease progression also indicated the benefit of the maintenance niraparib treatment. However, the OS analyses were limited due to missing data [56,57]. An FDA review of the updated OS data, presented in 2022, led to restrictions on the use of niraparib in second-line maintenance therapy for patients with g*BRCA* deleterious mutations. Final OS and long-term safety data from the study were presented at the 2023 Society of Gynecologic Oncology (SGO) Annual Meeting on Women’s Cancer, showing a numerically superior median OS in the g*BRCA*-mutant cohort of 40.9 months (95% CI: 34.9–52.9) compared to the placebo arm with 38.1 months (95% CI: 27.6–47.3). The superior OS above did not show up in between the g*BRCA*-mutant and the non-g*BRCA*-mutant cohort’s OS data [58]. Recently, in a real-world study among patients with *BRCA* wild-type recurrent ovarian cancer presented at the 2023 American Society of Clinical Oncology (ASCO) Annual Meeting, second-line maintenance niraparib therapy improved OS, compared to those with active surveillance only [59] (Table 2).

**Table 2 biomedicines-12-00593-t002:** List of clinical studies that have investigated somatic *BRCA* gene (s*BRCA*) mutation(s).

Clinical Trial	Phase	Treatment	s*BRCA*	g*BRCA*	Results
Vendrell et al. [43]	RCS	PARP inhibitor	*BRCA1*: *n* = 21 *BRCA2*: *n* = 12	*BRCA1*: *n* = 32 *BRCA2*: *n* = 15	Somatic variant showed a better outcome than germline (*p* = 0.049). Somatic alterations had longer survival than germline (5 y survival rate: 86.4% vs. 63.7%, *p* = 0.17).
ORZORA trial [47]	3	Olaparib	*n* = 55	*n* = 87	Similar clinical activity. Median PFS: - for s*BRCA*: 16.6 (95% CI: 12.4–22.2) - for g*BRCA* 19.3 (95% CI: 14.3–27.6)
ARIEL4 study [55]	3	rucaparib	*n* = 49	*n* = 275	median PFS for s*BRCA* 7.5 (5.6–11.2) months vs. 7.4 (7.3–9.2) months for g*BRCA*
ENGOT-OV16/NOVA [57,58]	3	niraparib	*n* = 47	*n* = 203	HRD-positive tumors and *BRCA* somatic mutation had a similar reduction in the risk of disease progression as that in the g*BRCA* cohort (PFS: 20.9 vs. 11.0 months; HR: 0.27; 95% CI: 0.08–0.90; *p* = 0.02)

g*BRCA*: germline *BRCA* gene mutation(s); HR: hazard rate; PFS: progression-free survival; RCS: retrospective cohort study.

Progression after first-line PARPi maintenance therapy is challenging, and several trials are currently exploring the PARPi rechallenge [60]. The next question is whether oligo progression during PARPi treatment can be effectively treated with a secondary surgical resection and continuous PARP inhibition instead of switching to chemotherapy.

In conclusion, the benefit of PARPi alone or in combination in the treatment of ovarian cancer is clear and recommended in the *BRCA* mutated or HRD-positive settings. Testing of the patients as early as possible, considering other factors, like platinum sensitivity and surgical outcome, and the results of the latest trials should be considered for up-to-date decision-making.

## 4. Breast Cancer

Women with germline pathogenic mutations in *BRCA1* or *BRCA2* have a significantly higher lifetime risk of developing cancers in several organs, especially in the breast and ovaries. Cumulative risk can be up to 57% (95% CI: 47–66%) and 49% (95% CI: 40–57%) for *BRCA1* and *BRCA2* mutations, respectively. Breast cancer cumulative risk is 60% and 55%, respectively. Based on these data, novel therapeutic approaches are eagerly awaited for this subgroup of patients. Additionally, it has to be mentioned that these genetic alterations are usually associated with other cancer types as well. Therefore, *BRCA*-targeted drugs could be used as general tumor-agnostic therapy as well [33,34].

BC is the most common cancer type in women, and below 1% of BC accounts for men. The risk of developing BC is higher if a positive family history is found. BC is a biologically and clinically heterogeneous disease, and patients with similar clinical stages have markedly different outcomes. Around 10% of patients have g*BRCA1/2* mutations, often leading to loss of function in genes implicated in DNA repair and cell cycle checkpoint activation. These patients are diagnosed with BC at a younger age, often with a positive family history of breast and/or ovarian cancer. Besides that, 90% of BC is caused by somatic mutations acquired lifelong; in *BRCA1/2* PV carriers, there is an increased lifetime risk of developing breast cancer. By the age of 80, those patients have up to 70% risk compared to 10% of the general population [61,62]. Extensive analyses have revealed that somatic *BRCA1* mutations are uncommon in unselected patients, but they can be important targetable mutations in metastatic disease [63]. Individuals with a g*BRCA1* mutation are more likely to develop triple-negative BC (TNBC) at a younger age. *BRCA1* mutation carriers develop predominantly but not exclusively estrogen receptor (ER) negative tumors, and there is an observation that patients with g*BRCA2* mutations are likely to develop ER-positive BC. g*BRCA* mutations are found in up to 23% of patients with TNBC and 5% of patients with ER-positive disease [62,64]. These patients are often diagnosed with locally advanced or metastatic disease, and despite aggressive chemotherapeutic regimens, they will relapse in a short time [65]. The lack of hormonal and human epidermal growth factor receptor 2 (HER2 receptor) in TNBC limited the possibility of an effective anticancer treatment. Nowadays, immunotherapy and PARPi can offer a better outcome. In BC, two PARPi monotherapies, Olaparib and talazoparib, have been approved by the FDA and the European Medicines Agency (EMA) for deleterious or suspected deleterious g*BRCA*-mutated HER2-negative BC, based on the positive outcomes of the phase III trials OlympiAD and EMBRACA [30,66]. The OlympiAD phase III trial for Olaparib in BC required a deleterious or suspected deleterious germline *BRCA* mutation as eligibility criteria [30]. The phase III trial for talazoparib in advanced BC by Litton et al. also included only g*BRCA1/2* mutations [66].

TNBCs often harbor somatic mutations or *BRCA* genes that may be silenced. Somatic *BRCA1/2* mutations are detectable in circulating cell-free DNA (cfDNA) in approximately 13.5% of patients with metastatic BC. In pre-clinical models, pathogenic somatic *BRCA1/2* mutations have been shown to respond to PARP inhibition [67]. The COMETA-breast trial was a proof-of-concept study enrolling heavily pretreated TNBC patients with centrally confirmed somatic *BRCA1/2* and no g*BRCA1/2* mutation. Olaparib did not show clinically or statistically significant antitumor activity [68]. In contrast, in the LUCY real-world study, the clinical effectiveness of Olaparib was confirmed for metastatic, HER2-negative BC, regardless of the ER expression level [69].

Olaparib Expanded, a phase II open-label, nonrandomized, investigator-initiated study, assessed Olaparib response in patients with metastatic BC with s*BRCA1/2* mutations or another g/s mutation in homologous recombination-related genes, which are non-*BRCA1/2*. Patients could either have had an s/g pathogenic or likely pathogenic variant of *BRCA1/2* or also germline or somatic alterations in one of DNA repair genes, such as *ATM*, *ATR*, *BAP1*, *BARD1*, *BLM*, *BRIP1*, *CHEK1*, *CHEK2*, *CDK12*, *FANCA*, *FANCC*, *FANCD2*, *FANCF*, *MRE11A*, *NBN*, *PALB2*, *RAD50*, *RAD51C*, *RAD51D*, or *WRN*. If s*BRCA1/2* was present, g*BRCA1/2* had to be excluded via germline testing. The objective response rate [ORR; patients have either partial (PR) or complete response (CR) to the treatment] was the primary endpoint, and there were further secondary endpoints such as clinical benefit rate and PFS. Confirmed responses were seen only with germline *PALB2* and s*BRCA1/2* mutations. With *ATM* or *CHEK2* mutations alone, no responses were observed. With Olaparib, the median ORR and PFS for germline *PALB2* and s*BRCA1/2* PV carriers were 82% and 13.3 months, and 50% and 6.2 months, respectively. These results are close to the ORR and median PFS with PARPi of 60% and 7–8.6 months reported in g*BRCA1/2* carriers in the OLYMPIAD and EMBRACA trials. The trial was the first report of PARPi response in patients with BC with somatic *BRCA1/2* mutations, and now Olaparib is a category 2b NCCN (National Comprehensive Cancer Network) guideline recommendation for treatment in metastatic disease, any subtype [70].

Similar results were reported in a single-institutional, retrospective study. Breast cancer patients with confirmed s*BRCA1/2* or g/s non-*BRCA* HRR mutations were included. Seven patients were treated with Olaparib, off-protocol, off-label for metastatic breast cancer. All s*BRCA1/2* mutation carriers responded to Olaparib, while other HRR-associated mutation carriers did not respond to PARP inhibition. Median PFS was 6.5 months with s*BRCA1/2* mutations, compared to the 3 months of those patients with non-*BRCA* HRR mutations [71]. However, the number of reported cases was low. The results have suggested that patients with tumors harboring s*BRCA1/2* mutations might benefit from the treatment with PARPi, similar to what we have seen in ovarian cancer. The identification of patients beyond g*BRCA1/2* carriers whose cancers may be sensitive to PARP inhibition is clinically meaningful. Similarly, more attention is needed for the hormone receptor-positive BC population, which represents 70% of *BRCA2*-associated BCs [72]. This significantly expands the population of patients with BC likely to benefit from PARPi treatment beyond those with g*BRCA* mutations, including those with subtypes other than TNBC (Table 3).

In conclusion, currently Olaparib and talazoparib are FDA-approved PARPis for BC patients with germline *BRCA* mutations. The use of PARPi therapies at early stages of BC and in patients without germline *BRCA* mutations are both subject to confirmation of PARPi efficacy in clinical trials.

## 5. Pancreatic Cancer

The incidence and mortality rates of pancreatic cancer patients are increasing nowadays. The amount of newly diagnosed cases has doubled since 1990 and is expected to become the second leading cause of cancer-related mortality in the next few years. About 5–9% of pancreatic cancer patients harbor germline PV of *BRCA* genes. Known g*BRCA* carrier individuals have a higher lifetime risk of developing pancreatic cancer compared to the normal population (with *BRCA1*: 2.2–3.0%; with *BRCA2*: 3.0–7.0%) [74,75,76]. In general, pancreatic cancer is diagnosed in an advanced stage, and despite personalized and modern treatments, it is associated with poor outcomes. In patients who undergo curative-intent surgery, the recurrence rate is relatively high despite early treatment. There appears to be no difference in actionable utility between g/s or *BRCA1/2* mutations. Promising and durable outcomes were observed in a subset of g/s *BRCA1/2* mutation carrier patients treated with platinum and PARP inhibitor therapies [77].

Olaparib is currently the only FDA-approved PARPi to treat pancreatic ductal carcinoma. However, only germline *BRCA*-mutated patients participated in the clinical trials. The POLO trial for pancreatic cancer, which studied Olaparib as maintenance therapy, did not include any somatic *BRCA* patients. This was the first study to show a biomarker-based treatment of pancreatic ductal carcinoma in patients with germline *BRCA* mutation. The study enrolled metastatic pancreatic cancer patients who had not progressed during first-line platinum-based chemotherapy. It is important to note that maintenance Olaparib treatment was compared to placebo, and the study concluded a PFS benefit but without OS advantage. The median PFS was 7.4 vs. 3.8 months [78].

Another PARPi, rucaparib was also evaluated in a maintenance setting after platinum treatment in a phase II study. Rucaparib was proven to be a safe and effective therapy for platinum-sensitive, advanced pancreatic cancer with a pathogenic variant in *BRCA1*, *BRCA2*, or *PALB2*. In this study, somatic *BRCA* patients were enrolled too. The findings of efficacy in patients with germline *PALB2* and s*BRCA2* mutations expand the population likely to benefit from PARP inhibition beyond g*BRCA1/2* variant carriers. The median PFS was 13.1 months, the median OS 23.5 months, and the ORR was 41.7% [79]. The RUCAPANC study was terminated early due to a lack of convincing results. Both germline and somatic *BRCA* patients were enrolled in this study, and the difference, compared to the studies above, was that platinum sensitivity was not included in the eligibility criteria [79] (Table 4). Indeed, data suggest benefits for the use of platinum-based systemic treatment in patients harboring *BRCA1/2* mutations in neoadjuvant or metastatic settings [77,80,81].

Studies with PARPi in this patient population have been mostly limited to patients with germline *BRCA* mutations. There are only a few publications with reported cases of s*BRCA*-mutated pancreatic cancers. Reiss et al. and Shroff et al. published small studies investigating the effect of rucaparib monotherapy, enrolling only one and three s*BRCA*-mutated patients, respectively; the pooled response was 75% for somatic *BRCA* [79,82] (Table 4).

There is an increasing need for combination treatments and novel therapeutic options in pancreatic cancer. Primary and acquired resistance to chemotherapy and PARPi is a challenge in improving long-term outcomes and maintaining quality of life. The use of PARPi in early and advanced stages and in combination with other novel therapies is now under investigation [83].

## 6. Prostate Cancer

Prostate cancer is the second most common cancer in men. Over the last decades, the development of targeted treatments has demonstrated improvement in OS and quality of life, too. Despite novel treatments, the disease remains fatal, and additional treatment approaches are needed. Systemic treatment recommendations, depending on stage, include androgen receptor (AR) signaling targeted therapy and chemotherapy as well [84]. However, germinal and/or somatic alterations of DNA damage response pathway genes are found in a substantial number of patients with advanced prostate cancers. Studies have suggested that PARP inhibition may provide benefits for patients with alteration in HR-related genes other than *BRCA1/2* without a clear view of which genes are consistently associated with response [85,86] (Table 5).

In the PROfound phase III trial [87], Olaparib was compared to hormonal therapy after progressing on at least one treatment with enzalutamide or abiraterone, with or without previous taxane chemotherapy, in patients with metastatic castration-resistant prostate cancer (mCRPC). Patients with an alteration in *BRCA1/2* or *ATM* were assigned to cohort A, and patients with other alterations were allocated to cohort B. PFS in the cohort A was longer in the Olaparib group (7.4 months vs. 3.6 months; HR: 0.34; 95% CI: 0.25–0.47; *p* < 0.001) [87]. Post hoc analysis of the subgroup of patients with mCRPC with *BRCA* alterations in the PROfound study has shown a PFS benefit with Olaparib in all zygosity subgroups. Patients with *BRCA2* homozygous deletions experienced prolonged responses to Olaparib (median radiological PFS: 16.6 months). It has to be noted that some evaluations of the study are limited by small patient numbers. For example, the germline DNA analysis was performed for 112 (70%) patients. The risk of disease progression was similar for patients with germline (*n* = 61; HR: 0.08; 95% CI: 0.03–0.18) and somatic (*n* = 51; HR: 0.16; 95% CI: 0.07–0.37) *BRCA* alterations [86]. The PROfound trial was the first study showing an improvement in OS for mCRPC with an alteration in *BRCA1/2* or *ATM* genes [88].

The FDA granted accelerated approval also to rucaparib in May 2020 for the treatment of adult patients with deleterious *BRCA* mutation (germline and/or somatic)-associated mCRPC who have been treated with androgen receptor-directed therapy and a taxane. This approval was based on data from the multicenter, open-label, single-arm TRITON2 trial [85]. Almost half of the TRITON2 patients with *BRCA*-mutated mCRPC had a complete or partial tumor size reduction with rucaparib. Clinical benefits were also observed with other DNA damage repair gene alterations. Later, the TRITON-3 randomized clinical trial of rucaparib was conducted in the pre-docetaxel setting. TRITON-3 was the second phase III trial to evaluate a PARPi in mCRPC after the PROfound trial of Olaparib and the first to compare a PARPi with docetaxel. The latter is the preferred treatment option for patients with metastatic disease who have progressed after an androgen receptor pathway inhibitor (ARPI). Rucaparib significantly improved radiographic PFS versus docetaxel or a second-generation ARPI in patients with *BRCA1/2*-mutated mCRPC. Two hundred seventy patients were assigned to receive rucaparib and 135 to receive a control medication. In the two groups, 201 patients and 101 patients, respectively, had a *BRCA* alteration. At 62 months, the duration of imaging-based PFS was significantly longer in the rucaparib group compared to that of the control group. In the *BRCA* subgroup, the median PFS was 11.2 months vs. 6.4 months (HR: 0.50; 95% CI: 0.36–0.69; *p* < 0.001), while in the intention-to-treat group, it was 10.2 months vs. 6.4 months (HR: 0.61; 95% CI: 0.47–0.80; *p* < 0.001). In an exploratory analysis in the *ATM* subgroup, the median duration of imaging-based PFS was 8.1 months in the rucaparib group and 6.8 months in the control group (HR: 0.95; 95% CI: 0.59–1.52) [89] (Table 5).

There are three known currently recruiting trials in prostatic cancer with *BRCA* alterations. In two of them, the administered drug is a PARPi. In detail: (1) Pamiparib is given in castration-resistant mCRPC patients with s/g*BRCA* mutation or HR-deficiency [90]. (2) In the NePtune trial, Olaparib is given in a neoadjuvant setting with LHRH-agonist for prostatic cancer patients with *BRCA* alterations and high-risk or unfavorable intermediate-risk tumors [91]. (3) There is a study about CX-5461 for patients with s*BRCA2* mutations and/or *PALB2* mutations in the pancreas/prostate/breast or ovary malignancy with the corresponding g*BRCA2*/*PALB2* mutation [92].

**Table 5 biomedicines-12-00593-t005:** Summary of studies where prostate cancer patients with somatic *BRCA* mutated (s*BRCA*) cancers were treated with PARP inhibitors.

Clinical Trial	Phase	Treatment	s*BRCA*	g*BRCA*	Results
PROFOUND trial [86]	3	Olaparib	*n* = 51	*n* = 61	Radiographic PFS benefit was investigated. Risk of disease progression was similar for patients with: - g*BRCA*: HR: 0.08 (95% CI: 0.03–0.18) - s*BRCA*: HR: 0.16 (95% CI: 0.07–0.37)
TRITON 2 study [85]	3	rucaparib	+	+	ORR was similar between patients with a g*BRCA* and s*BRCA* alterations
TRITON 3 study [89]	3	rucaparib	+	+	Median PFS in the *BRCA* subgroup was 11.2 months for rucaparib vs. 6.4 month

g*BRCA*: germline *BRCA* mutation; ORR: objective response rate; PFS: progression-free survival.

## 7. Conclusions

This review aimed to summarize the clinical knowledge and discuss further possibilities about the role and use of PARP inhibition in somatic *BRCA* mutations across different tumor types. The predictive role of somatic *BRCA* mutations is not entirely clarified. Since *BRCA1* was discovered in 1994, most of the clinically relevant studies and reports have been limited to germline mutations and their importance and care [2]. Nowadays, there is accumulating evidence for routine somatic *BRCA* mutation testing, but the relevance of *BRCA* epigenetic modifications is less clear [68]. Targeted therapeutic agents such as PARP inhibitors have emerged in treating certain cancers associated with germline *BRCA* mutations. In the last decade, studies have demonstrated the effectiveness of PARP inhibitors in treating patients with somatic *BRCA* mutations as well; however, overall, the number of treated somatic *BRCA* mutation patients is still low. However, in the case of some tumors, there are an increasing number of studies involving s*BRCA* cases, such as ovarian carcinoma [45,48,49,50,51].

Patient access to next-generation sequencing and detection of actionable mutations is needed for more precise and optimal treatment selection. Comprehensive genomic alteration testing may provide novel clinical strategies for personalized therapy in advanced tumors with improvement in OS and quality of life. More trials regarding molecular targeted therapy are expected to be conducted in the future, and at the same time, mechanisms regarding resistance are expected to be explored and understood, which will aid the development of strategies to resensitize tumor cells to PARPi and improve long-term effectiveness.

Since the first approval of Olaparib in 2014, PARPis have been used in oncological care, mainly for patients with breast, ovarian, and prostate cancers. The PARPi class is generally well-tolerated, and oral administration is also preferred. Every agent possesses a unique side-effect profile, and treatment with the different PARPis may result in various adverse events and should not be considered one entity. Among the most common adverse events associated with almost all PARPi treatments in clinical trials were fatigue, anemia, neutropenia, thrombocytopenia, nausea, vomiting, diarrhea, headache, and decreased appetite [93]. Niraparib and talazoparib have more prominent hematologic adverse event profiles, while niraparib has an increased risk of cardiac events [94]. Compared to combined chemotherapeutic treatments, the PARPis are well tolerated and ensure a better quality of life.

Olaparib is currently approved as monotherapy for the maintenance treatment of adult patients with platinum-sensitive relapsed *BRCA*-mutated, germline and/or somatic, high-grade serous epithelial ovarian, fallopian tube, or primary peritoneal cancer who are in complete response or partial response to platinum-based chemotherapy. In an early phase study, 19 and 20 patients were identified with a somatic tumor *BRCA* mutation. The limited data for these s*BRCA* mutated patients show that fewer patients on Olaparib reported progression events or death events compared with placebo [95,96]. Olaparib is also indicated in combination with bevacizumab for the maintenance treatment of patients with advanced high-grade epithelial ovarian cancer following first-line platinum-based chemotherapy and whose cancer is associated with homologous recombination deficiency positive status defined by either a *BRCA1/2* mutation and/or genomic instability.

BC treatment is no longer so simple, as there is a lack of studies showing a convincing benefit for the treatment of s*BRCA* mutated patients. Olaparib is approved for the adjuvant, locally advanced, or metastatic treatment of HER2-negative BC only in case of g*BRCA* mutational status. However, based on a recently published phase II study, the use of Olaparib in s*BRCA* mutated tumors is a promising possibility [70]. Moreover, Olaparib is an indication for prostate cancer as a monotherapy or in combination with abiraterone and steroids for the treatment of patients with metastatic disease whose chemotherapy is not clinically indicated. Both germline and/or somatic *BRCA* mutation status are accepted and recommended by guidelines.

Talazoparib was first approved for HER2-negative, *BRCA*-mutated, locally advanced, or metastatic BC. A phase II trial evaluating the efficacy of talazoparib for somatic *BRCA*-mutant, HER2-negative metastatic BC is ongoing [97], and its results are eagerly awaited. Recently, talazoparib was announced by FDA as the third PARPi to treat HRR gene-mutated mCRPC, based on the randomized phase III clinical trial TALAPRO-2 [98].

Niraparib, veliparib, and rucaparib are indicated as monotherapy for the maintenance treatment of first-line and recurrent ovarian, fallopian tube, or primary peritoneal cancer. FDA review of updated OS in 2022 led the regulatory agency to restrict niraparib’s second-line maintenance indication to patients harboring deleterious or suspected deleterious germline *BRCA* mutations. In the US, rucaparib is also indicated for the treatment of adult patients with a deleterious *BRCA* mutation germline and/or somatic mCRPC who have been treated with androgen receptor-directed therapy and taxane-based chemotherapy.

A meta-analysis comparing the ORR of PARPi in patients harboring somatic versus germline *BRCA* mutations was published by Mohyuddin et al. in 2020 [99]. Although the search strategy was limited to randomized trials and cohort studies enrolling both somatic and germline *BRCA* patients, across 18 studies, a total of 236 patients with somatic *BRCA* mutations were identified, and ORR and PFS were compared to g*BRCA* data. Somatic *BRCA* mutations were defined as either a *BRCA1* or *BRCA2* mutation present only in the tumor tissue. In those studies, 24 out of 43 patients with somatic *BRCA* mutations (55.8%) and 69 out of 157 (43.9%) patients with germline *BRCA* patients had a response to therapy due to PARPi. This difference was not statistically significant (*p* = 0.399). In all five studies that reported PFS, there was no obvious difference in the outcomes between somatic versus germline *BRCA* patients. However, a precise statistical analysis could have not been performed. A subgroup analysis accounting for different malignancy types and different PARPis also did not reveal any significant difference in response rates [99]. What is particularly important, given the increasing number of breast cancer patients, are clinical trials enrolling somatic *BRCA* mutations. Now, there are two ongoing observational studies in breast cancer patients, but no results have been posted yet [100,101]. In the RADIOLA study, the RAD51-foci score is investigated if it has a predictive value on Olaparib efficacy in g/s*BRCA* or *PALB2* or *RAD51C/D* mutation carrier advanced breast cancer patients; no results were posted [102].

In conclusion, our knowledge of somatic mutations in tumors could provide insights into tumorigenesis and reveal candidates for targeted therapeutics. The identification of patients beyond g*BRCA1/2* carriers whose cancers may be sensitive to PARPi remains an important goal. Not only in early disease but especially those patients suffering from metastatic disease need targeted therapies to improve survival and maintain quality of life.

## Figures and Tables

**Figure 1 biomedicines-12-00593-f001:**
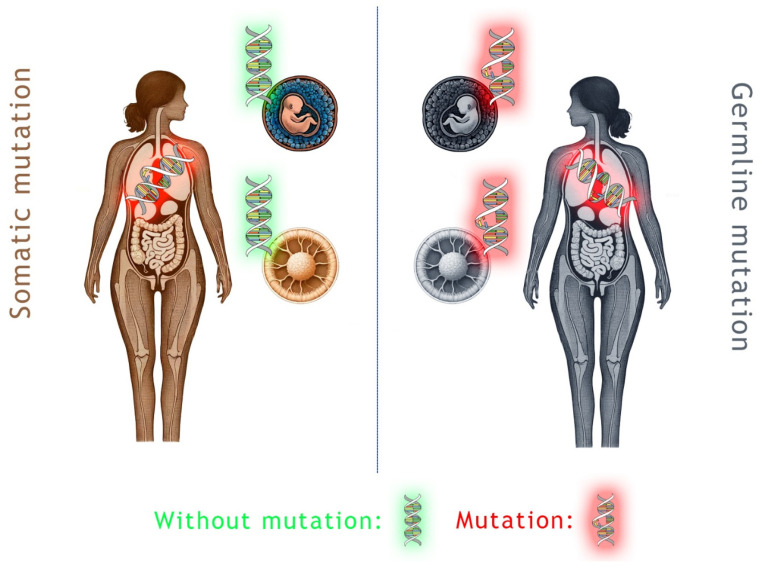
Schematic representation of germline and somatic *BRCA* gene mutations. In the case of germline mutation, all cells of the body contain the mutated gene variant, and it is always passed on from parent to offspring. Somatic mutations occur at a later stage of ontogeny, and the random mutations develop during normal mitotic cell divisions. Note: The schematic human figure, the embryo, and the oocyte were created with the assistance of DALL·E 2; otherwise, the authors did every other aspect of the figure, including the different coloring, arrangement, and labeling. The normal and mutated DNA sequence is courtesy of Wikimedia Commons; the unmodified original was created by NASA/David Herring and distributed under the CC0 1.0 license.

**Table 1 biomedicines-12-00593-t001:** Clinical differences of somatic (s*BRCA*) and germline *BRCA* (g*BRCA*) mutations.

Comparison	g*BRCA* Mutation	s*BRCA* Mutation
Detection technique(s)	Peripheral blood test for known hereditary pathogenic mutations	NGS testing of tumor tissue or Peripheral blood test for known circulating tumor DNA markers
Mutation stability	Constant	Changes with time and tumor progression
Testing criteria	Early age at tumor diagnosis	Tumor profiling clinically indicated for treatment actionability and prognosis
Risk factor(s)	Positive family history	Ø
Clinical management:		
- Cancer patient	Various management and treatment options	
- Family members	Screening and preventive methods	No action needed
Management	Genetic counseling strongly recommended pre- and post-detection	Oncological board discussion

NGS: next-generation sequencing.

**Table 3 biomedicines-12-00593-t003:** Summary of studies that have investigated breast cancer patients with somatic *BRCA* mutated (s*BRCA*) tumors who were treated with PARP inhibitors.

Clinical Trial	Phase	Treatment	s*BRCA*	g*BRCA*	Results
Olaparib expanded [70]	2	Olaparib	*n* = 16	*n* = 0	ORR for s*BRCA1/2* carriers was 50%. Median PFS for s*BRCA1/2* carriers was 6.2 months.
COMETAbreast study [68]	2	Olaparib	*n* = 11	*n* = 0	Heavily pre-treated patients, no clinically or statistically significant antitumor activity. Median PFS: 2 months (95% CI: 1–4) Median OS: 9 months (95% CI: 1–14)
LUCY study [69]	3b	Olaparib	*n* = 3	*n* = 253	Small patient number for s*BRCA*, limited assessment of the clinical effectiveness.
Walsh et al. [71]	rObs	Olaparib	*n* = 4	+	Median PFS: - s*BRCA1/2*: 6.5 months (range 5–9) - non-*BRCA1/2*/HRR: 3 months (range 2–4)
RUBY trial [73]	2	rucaparib	*n* = 4	+	s*BRCA1*: 1 SD s*BRCA2*: 1 PR

g*BRCA*: germline *BRCA* mutation; ORR: objective response rate; OS: overall survival; PFS: progression-free survival; PR: partial response; rObs: a single institution, retrospective observational study; SD: stable disease.

**Table 4 biomedicines-12-00593-t004:** Summary of studies where pancreatic adenocarcinoma patients with somatic *BRCA* mutations (s*BRCA*) were treated with PARP inhibitors.

Clinical Trial	Phase	Treatment	s*BRCA*	g*BRCA*	Results
Reiss et al. [79]	2	rucaparib	*n* = 2	*n* = 34	PFS 13.1 month; OS 23.5 month; ORR 41.7% Median response: 17.3 months (95% CI: 8.8–25.8). Responses occurred in ½ of the s*BRCA2* (50%) and in 11/27 of the g*BRCA2* (41%) patients.
Shroff et al. [82]	2	rucaparib	*n* = 3	*n* = 16	ORR: 2/3 (67%) for s*BRCA* and 1/16 (6%) for g*BRCA*.

g*BRCA*: germline *BRCA* mutation; ORR: objective response rate; OS: overall survival; PFS: progression-free survival.

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
