# Peer review of "Predictive Value and Therapeutic Significance of Somatic BRCA Mutation in Solid Tumors"

_biomedicines, 2024, doi:10.3390/biomedicines12030593_

Round 1
Reviewer 1 Report
Comments and Suggestions for Authors
This review aimed to summarize and discuss further possibilities about the role and use of PARPi treatment in somatic BRCA mutations across different tumor types. I suggest improving the manuscript's readability if the data related to the OS efficacy of different treatments are reported in a table with related references.
Also, the discussion needs to be improved considering the possible clinical strategy to improve OS in different treatments which is not debated or discussed in the present form. Do the biomarkers of inflammatory reactions have a role in the long-term effects of the PARPi like for monoclonal mabs ?? please explain
see for references
Inflammatory Related Reactions in Humans and in Canine Breast Cancers, A Spontaneous Animal Model of Disease.
Front Pharmacol. 2022 Feb 11;13:752098. doi: 10.3389/fphar.2022.752098. eCollection 2022.Author Response
Please see the attachment.

Reviewer 2 Report
Comments and Suggestions for Authors
The article titled " Predictive value and therapeutic significance of somatic BRCA 2
mutation in solid tumors" has been evaluated and it requires some improvements to attract readers. The authors need to include more recent articles and connect different studies to make the article more informative. There are major concerns that need to be addressed.
1. The number of references provided is not sufficient, and the authors should include recent studies to elaborate on their points.
2. The article should be written in a way that captures the general audience's attention, not just experts in the field.
3. The authors should focus on making the figures interesting and engaging enough to encourage readers to finish reading the article or at least go through the figures to learn something new.
4. The authors need to consider how the readers can get the most out of the figures without reading the text in detail. It is crucial to find ways to maintain the readers' interest after two-thirds of the figures.
5. The authors should abbreviate BRCA1 and BRCA2 terms. Also, include a list of abbreviations at the end of the conclusion.
6. The authors should include the Tables for a better presentation.
Comments on the Quality of English LanguageModerate editing of English language required
Round 2
Reviewer 1 Report
Comments and Suggestions for Authors
The review is largely improved
Reviewer 2 Report
Comments and Suggestions for Authors
The authors performed significant revisions based on the reviewer's comments, the MS can be acceptable for publication.